# Productive Yield, Composition and Nutritional Value of Housefly Larva Meal Reared in High-Altitude Andean Zones of Peru

**DOI:** 10.3390/ani15142054

**Published:** 2025-07-11

**Authors:** Isai Ochoa, Emperatriz Valderrama, Elisa M. Ayquipa, Ludwing A. Cárdenas, Delmer Zea, Zenaida Huamani, Giorgio Castellaro

**Affiliations:** 1Faculty of Veterinary Medicine and Zootechnics, National University Micaela Bastidas of Apurimac, Abancay 03000, Perulcardenas@unamba.edu.pe (L.A.C.); dzea@unamba.edu.pe (D.Z.); zhuamanih@unamba.edu.pe (Z.H.); 2Department of Animal Production, Faculty of Agricultural Sciences, University of Chile, Santiago 8820808, Chile; gicastel@uchile.cl

**Keywords:** Diptera, insect meal, animal feed

## Abstract

Animal production in Peru is highly dependent on protein inputs such as soybean meal and fishmeal and is considerably affected when the price of these inputs increases. Diptera and their larval forms have been successfully tested in the feeding of poultry, fish and other domestic animals and have become an alternative as a protein input for the formulation and preparation of rations for farm animals. Therefore, this study evaluated the productivity, nutritional composition, amino acid profile, fatty acid profile and presence of *Salmonella* spp. in housefly larva meal reared in high Andean zones of Peru. In rearing conditions, the type of organic substrate influences the growth and productivity of the larva. Regarding its nutritional composition, it was determined that the larva meal contains an adequate balance of nutrients and does not have the presence of *Salmonella* spp.

## 1. Introduction

The forecast of world population growth indicates that the pressure on the environment for food production will increase [1], being one of the most important global challenges to achieve food security that is sustainable with the environment [2]. In this context, insect farming is a sustainable alternative for food production for humans and animals in the coming years [3], since they contain high concentrations of protein, lipids, vitamins and minerals and their production is carried out with various organic wastes, which reduce environmental pollution and give added value to these products [2], improving the management of organic wastes and generating new sources of income for producers [4].

The idea of using dipteran larvae to process organic waste was proposed almost 100 years ago. Since then, numerous laboratory studies have shown that several species of flies are well suited for the biodegradation of organic wastes, with the housefly (*Musca domestica*) and the black soldier fly (*Hermetia illucens*) being the most studied insects for this purpose [5]. It has been shown that the housefly (HF) and black soldier fly (BSF) can be used for the management of dairy, poultry and swine manure being an attractive solution to reduce the application of raw manure contaminants and other risks associated with the storage of raw manure [6]. The crude protein content of dipteran larval forms is high(42–63%) and so are the lipid contents (up to 36%), which may possibly be extracted and used for various applications, including biodiesel production [7].

Studies have shown that housefly larva meal (HFLM) provides nutrients for ruminant nutrition, in addition to rapid digestibility, which allows reproduction with a high nutritional content due to the levels of protein, lipids and minerals [8]. It is reported that the substitution of soybean meal with housefly larva meal did not affect the feed digestibility and growth of poultry [9]. Similarly, the inclusion of black soldier fly larva meal (BSFLM) in guinea pigs’ (*Cavia porcellus*) diets did not affect their meat quality [10]. Tilapia raised with larvae produced in poultry manure showed better growth performance and higher nutrient concentration, complying with the maximum levels of contaminants in food established by the WHO [11]. Larval forms of the housefly have also been successfully included in fish, poultry, and swine feed [3,7].

Housefly larvae can be reared on organic substrates such as poultry manure, pig manure, cattle blood with wheat bran, cattle blood and intestinal contents, fish guts and waste mixtures [7], being also successfully produced in cattle manure [6] and food waste mixtures [11]. Previous studies have reported that HFLM produced in a mixture of blood and wheat bran contained on average 47% crude protein, 25.3% fat, 7.5% crude fiber and 6.25% ash [12]. When larvae are produced with animal manure, their protein composition is higher than 50%, with high levels of Ca (1.32 to 1.47%) and P (1.72 to 2.09%) [13]. It has been documented that the amino acid profile of HFLM contains nine essential amino acids, highlighting lysine and methionine (6.04% and 2.28%, respectively) [11]. It is also important to mention that new techniques are being developed, among them encapsulation, which gives the product a more pleasant appearance and odor compared to typical insect flours, to improve its acceptance by the population [14].

It has been demonstrated that the production of larval forms of flies in pilot plants and on a large scale are effective and economically viable, since it allows the production of a food input, in addition to reducing greenhouse gas emissions [5]. Likewise, methane reduction was evident in a study with BSF larvae and HF larvae, minimizing carbon levels in animal production [8].

In Peru, the regions of Apurímac, Ayacucho and Huancavelica are those most affected by desertification and drought, which significantly affects human development indexes and is reflected in higher levels of poverty and extreme poverty [15]. The high-altitude Andean zones of Peru are characterized by their economic activity, which is predominantly agricultural, with cattle, sheep, guinea pigs, poultry and pigs being the mainstay of the agricultural units [16]. In these agricultural units, animal manure is not processed and is used as raw fertilizer during the planting season, which does not allow adequate use of these resources. The production of larval forms of houseflies is an opportunity to promote the circular economy in agricultural units, since a source of nutrients for animals may be obtained and the disposal of animal manure for crop fertilization may be improved.

Based on the above, the present study has evaluated the productivity and nutritional composition of housefly larvae reared on manure mixtures from different domestic animals in Andean highland areas of Peru.

## 2. Materials and Methods

### 2.1. Study Area

This study was conducted at the dipteran production center of K-Vet S.R.L., located in the Toraya district, Aymaraes province, Apurímac region, Peru (Figure 1). The district is located at an altitude of 3146 mASL, it has a maximum temperature of 21 °C and a minimum temperature of 5 °C, with an average rainfall—between September and March—of 13 mm per month [15].

### 2.2. Methods

This study was developed in two stages. The first stage evaluated the effect of the type of manure on larval productivity and the second stage evaluated the nutritional composition, amino acid profile, fatty acid profile and presence of *Salmonella* spp. of the HFLM.

#### 2.2.1. First Stage

In Peru, there is no governmental institution or legislation that regulates or restricts the collection of houseflies, as this species is neither protected nor considered endangered. For this study, only a limited number of adult flies were captured to establish the initial breeding stock under controlled conditions.

The adult flies were captured in the wild in the Toraya district, then transferred to the production center of Diptera of the company K.Vet S.R.L. (Abancay, Peru), which consisted of an airtight room of 5 × 3 m, with a photoperiod of 12 h light, with an average temperature of 25 ± 3 °C and a relative humidity between 50 and 60% and data was measured by Digital Thermohygrometer (Thermohygrometer UNI-T UT333, Dongguan, Guangdong Province, China).

The flies feed consisted of a mixture of 80 g of brown sugar, 250 mL of pasteurized milk and distilled water [17]; the food was placed daily at 8:00 in 4 flat plastic dishes with a diameter of 16 cm × 2 cm in height.

For oviposition, plastic containers of 250 mL capacity were used where enough cotton was placed to avoid compaction [17]. The containers were soaked with pasteurized milk, so that the flies could deposit the eggs. Once the eggs hatched, 200 live larvae were selected and transferred to the rearing medium, which consisted of a high-density polyethylene culture tray with 500 g of manure. In this first stage, the effect of the type of manure on the productivity of housefly larvae was evaluated using 3 types of manure and 3 combinations. The types of manure selected were those that can be easily obtained from the farms of the high Andean zones of the Apurimac region, being T1: 100% pig manure; T2: 100% guinea pig manure, T3: 100% poultry manure; T4: mixture of 50% guinea pig manure + 50% poultry manure; T5: mixture of 50% guinea pig manure + 50% pig manure; T6: 50% poultry manure + 50% pig 135 manure. Dried and ground manure was used and mixed according to the type of treatment.

The experimental unit consisted of a high-density polyethylene culture tray, measuring 30 × 20 × 4 cm, containing 500 g of fresh manure according to treatment, in which 200 larvae were inoculated. Ten replicates per treatment were used, giving a total of 60 trays.

Housefly larvae were harvested according to their growth, with the yellowish color of the larvae being an indicator for the harvest. Wooden sieves with 4 mm galvanized mesh were used to separate the larvae from the culture substrate. Negative phototropism was then used to separate the larvae from the remaining substrate.

To evaluate the weight of the larvae, an analytical weighing balance (OHAUS^®^, Parsippany, NJ, USA) was used, which has a range of 0–250 g and an accuracy of 0.001 g. To measure larval size, a Digital Vernier Caliper with a range of 1 mm was used. To evaluate the percentage of larval mortality, the difference between larvae that were inoculated and the number of larvae harvested was calculated.$$\%\;de\;mortality=\frac{Dead\;larvae\;}{Cultured\;larvae}\;\times 100$$

#### 2.2.2. Second Stage

The second stage of this study consisted of producing housefly larvae and evaluating their nutritional composition, amino acid profile, fatty acid profile, in vitro digestibility and the presence of *Salmonella* spp.

The production of larvae was conducted considering the results of the first stage, in which it was demonstrated that greater productivity can be obtained using poultry manure combined with pig manure in a proportion of 50%.

For fly feed, the same procedure as in the first stage was used. The flies had access to the production boxes, which consisted of expanded polystyrene thermal boxes that offer high protection capacity and isothermal insulation with dimensions of 51 × 76 × 58 cm, where 1000 g of the manure mixture was added every 24 h until larval development was complete.

On day 10, larvae were harvested using the method described in the first stage. The harvested larvae were subjected to dehydration, using an oven with a forced air circulation system (Memmert^®^, Schwabach, Germany), at a temperature of 70 °C, for 5 h until a humidity of approximately 10% was obtained.

Grinding was conducted in the processing area of K-Vet feed, using a hammer mill model MMV-06, with a grinding capacity of 100 kg/h. The HFLM obtained from the milling process was hermetically packed in high density polyethylene bags and stored in a dry environment, with the identification of the production lot.

The samples used in nutritional and microbiological analysis were collected in high-density polyethylene bags with a capacity of 500 g, each containing a 350 g sample. A total of 10 random samples were taken, each from a different production batch of HFLM, ensuring representative sampling across 10 distinct lots. All samples were stored at −20 °C until analysis. Chemical analysis of HFLM was performed using the methods described in Table 1.

**Table 1 animals-15-02054-t001:** Methods of analysis of the components of housefly larva meal.

Analysis	Method	Reference
Total protein	Micro Kjeldahl Method	AOAC 954.01 [18]
Dry Matter (Moisture)	Drying of raw material at 105 °C for 24 h	AOAC 934.39 [18]
Ashes	Calcination in muffle at 600 °C	AOAC 942.05 [18]
Ethereal extract	SOXHLET Method	AOAC 920.39 [18]
Crude fiber	Determination of crude fiber	NTP 205.003 2016 [19]

Ten samples of 250 g each were sent for fatty acid profile analysis, which was conducted using the method described by Li and Watkins (2001), titled “Analysis of fatty acids in food lipids” [20].

Ten samples of 100 g each were analyzed for the detection of *Salmonella* spp., following the method described by the International Commission on Microbiological Specifications for Foods (ICMSF-2000) [21]. *Salmonella* spp. was selected for microbiological analysis because, in Peru, national food safety regulations, specifically the Technical Health Standard NTS N° 071-MINSA/DIGESA-V.01 [22], require that food and feed products must be free from *Salmonella* contamination. This pathogen is considered a critical indicator of sanitary quality and food safety, particularly in both human consumption and animal production systems [22].

Total energy content (MJ/kg) was determined by calorimetry (Parr^®^ 6200 Isoperibol Calorimeter, Moline, IL, USA)

Five samples of 200 g each were used for the analysis. In vitro protein digestibility was determined using the feed and forage analysis method described by Becker (1961) [23], which simulates gastric digestion using a 0.0002% pepsin solution in 0.075 N hydrochloric acid. Defatted and dried samples were incubated at 45 °C for 16 h. A control sample without pepsin was included in all cases. After incubation, the reaction mixture was filtered using Whatman No. 2 filter paper to retain the insoluble fraction. The crude protein content of the residue was determined using the Micro Kjeldahl method. Digestibility was calculated using the following formula:$$\%Digestibility=\left(\frac{g\;of\;residual\;protein\;without\;pepsin-g\;of\;residual\;protein\;with\;pepsin}{g\;of\;residual\;protein\;without\;pepsin}\right)\;\times 100$$

A 250 g sample was used to evaluate the amino acid composition of HFLM. The analysis was performed by reversed-phase high-performance liquid chromatography as described in Analytical Biochemistry, 1984 [24].

The amino acid score was determined based on the essential amino acid content, using the requirements recommended for humans by the Food and Agriculture Organization of the United Nations (FAO) [25], for which the following formula was used:$$Amino\;acid\;score=\frac{Test\;amino\;acid}{Reference\;amino\;acid}\times 100$$

### 2.3. Statistical Analysis

For the data analysis of the growth and development of housefly larvae, a general linear fixed-effects model was used [26], whose general expression was as follows:*Y_ijk_* = *μ* + *TS_i_* + *e_ijk_*(1)

In the above equation, *Y_ijk_* represents the variable under analysis (larval weight, larval size); *TS_i_*, the fixed effect of the type of organic substrate (poultry manure, pig, guinea pig and combinations); and *e_ijk_*, is the experimental error. The assumptions of the analysis of variance were evaluated before statistical analysis. Conversion of larval mortality data was performed by angular transformation. When there was a difference between treatments, means were separated by Tukey’s multiple comparison test with a *p* value of 5%.

Larval mortality was analyzed using the Kruskal–Wallis test, due to non-compliance with ANOVA assumptions.

The data of nutritional composition and fatty acid profile of the HFLM were analyzed by descriptive statistics, calculating the mean and standard deviation. In addition, confidence intervals were used to represent the results. To perform the above analyses, InfoStat software version 2020 was used [26].

## 3. Results

### 3.1. Larval Development According to Culture Substrate

As shown in Table 2, the type of manure used as feed to produce housefly larvae significantly affected the final weight (*p* < 0.0001), total weight gain (*p* < 0.0001) and larval size (*p* < 0.0001) at the end of larval development.

Regarding final weight and total weight gain, the larvae with the greatest increase in these variables were those larvae produced in combination of poultry manure (50%) and swine manure (50%), achieving a final weight of 27.60 ± 0.58 mg and a total weight gain of 24.58 ± 0.59 mg. The treatments in which lower final weight and lower total weight gain were reported were guinea pig manure (13.16 ± 0.71 mg and 10.11 ± 0.71 mg, respectively) and the combination of guinea pig manure with poultry manure (13.29 ± 0.71 mg and 10.29 ± 0.71 mg).

The larvae with the largest size were those produced in the combination of poultry manure (50%) and pig manure (50%), obtaining a size of 9.47 ± 0.16 mm at the time of the harvest. The larvae with the lowest growth were those grown in guinea pig manure, which at the time of the harvest obtained a size of 3.79 ± 0.05 mm.

Table 3 describes the larval development time and the percentage of larval mortality according to the organic substrate used in larval production. The analysis of variance shows that for larval development time, there were significant differences (*p* = 0.0001) between the different types of manure, with guinea pig manure being the substrate that takes the longest time to develop larvae. The Kruskal–Wallis test indicates that there are significant differences (*p* = 0.001) for the percentage of mortality, with higher mortality in pig manure with 3.85 ± 0.62% and lower mortality in larvae produced in poultry manure and mixtures.

### 3.2. Nutritional Composition, In Vitro Digestibility and Energy Content of Housefly Larvae Meal

Table 4 shows the results obtained in the analysis of the nutritional composition, in vitro protein digestibility and total energy content of HFLM on a dry matter basis. The high protein content (56.50%) and ethereal extract (13.07%) with an energy contribution of 16.36 MJ kg^−1^.

### 3.3. Amino Acid Profile of Larval Meal

Table 5 presents the results of the amino acid composition analysis of the HFLM, highlighting its higher content of glutamic acid, aspartic acid, threonine and arginine. The amino acids with the lowest content were tryptophan and methionine. Similarly, Table 5 describes the essential amino acid scoring patterns for infants, children, adolescents and adults (modified values from the 2007 WHO/FAO/UNU report) [25]. Finally, the amino acid score of larva meal for essential amino acids is presented, showing a deficit for isoleucine, leucine and methionine.

The essential amino acids in HFLM (threonine, valine, methionine, leucine, phenylalanine, isoleucine, cysteine, lysine) represent approximately 37.47% of the total amino acids and many of them conform to the requirements for amino acids recommended by the Food and Agriculture Organization of the United Nations (FAO) [25], so HFLM can be considered as a good source of protein for animal feed formulation.

### 3.4. Fatty Acid Profile of Housefly Larvae Meal

Table 6 shows the results for the saturated and unsaturated fatty acid content of the HFLM, predominantly palmitic acid with 29.34%, palmitoleic acid with 21.65% and cis-9 oleic acid with 26.53%.

### 3.5. Evaluation of Salmonella *spp*. Content

The presence of *Salmonella* spp. in the housefly larva meal samples was evaluated, and the absence of *Salmonella* spp. was reported in all samples evaluated.

## 4. Discussion

### 4.1. Larval Development According to Type of Manure

This study evaluated how different types of animal manure affect the growth of HF larvae (weight gain, final size, and larval mortality). Regarding larval weight, larvae had a total weight gain between 10.11 ± 0.71 mg with guinea pig manure and 24.58 ± 0.59 mg when using a mixture of poultry and pig manure. Our results agree with those reported by Larrain and Salas [27], in which housefly larvae had weights between 8.4 mg with horse manure and 20.0 mg with pig manure. In larvae reared with bovine manure mixed with wheat bran and bovine blood, larvae weighed between 17 and 17.7 mg with larval sizes between 8.7 and 9.8 mm [28]. On the other hand, it has been reported that the type of substrate and its interaction with egg load have a significant effect on egg viability and larval productivity [29].

In contrast to what was reported in our study, it is indicated that the maximum larval weight was reached 4 days after inoculation, and the percentage of survival to the pupal stage varied, being the highest in swine manure (73%), and poultry manure (67%), while 50% survived when fed with dairy cow manure [6]. In diets with high starch content, a higher proportion of larval survival was observed, obtaining 80% survival with 0% starch and 18% with substrates up to 50% starch.

It has been described that nitrogen is a key nutrient affecting the digestion of housefly larvae [6], the conversion into a larval biomass of protein-poor fractions could be increased and the time of the treatment process decreased by the addition of a protein-rich substrate, while a higher utilization of available nutrients could be expected by combining a readily available carbon-rich substrate with a protein-rich substrate [30]. The the high proportion of cellulose hemicellulose and lignin in manure decreases biomass conversion, with an increase in larval biomass being observed with the addition of poultry manure [4].

Zhu et al. [31] indicates that fresh pig manure can be adequately composted for larval production, ensuring larval production of at least 100 kg/ton in 7 days, which can be used as a feed supplement and residual compost as organic fertilizer. In a study conducted in Namibia, using pig manure, poultry manure and a mixture of both, it was determined that pig manure produced the highest larval yield and housefly larvae were shown to reduce the organic matter used by 40–50%, making the organic waste easier to handle, store and potentially be used as soil fertilizer for crop production [13]. Housefly larvae reared on fishery waste achieved the highest larval weight and fat content being a suitable alternative for larval production [32,33].

From what has been described in the various investigations, the development of housefly larvae would be associated with the diet of the animals from which the organic substrate comes, since poultry and pig manure contains a higher percentage of nitrogenous matter than the diet of purely herbivorous animals in which the fiber content predominates. Although guinea pig manure is produced in most agricultural units in high Andean zones, to achieve greater larval productivity it should be mixed with manure that provides more nitrogenous compounds such as pig and poultry manure, which would allow the application of the circular economy in these productive sectors.

### 4.2. Nutritional Composition of Housefly Larvae Meal

According to the results, HFLM can be classified as an input of high protein value (56.5% crude protein), with a high fat content (13.07%) and an adequate contribution of soluble carbohydrates (12.03%). In addition, it is important to highlight its low crude fiber level (6.77%), which would be related to its high protein digestibility (88.3%). The results are consistent with those reported by Makkar et al. [7] who in a meta-analysis of existing information up to 2014, reported that housefly larvae contain on average 50.4 ± 5.3% protein, 5.7 ± 2.4 crude fiber, 18.9 ± 5.6 lipids and 10.1 ± 3.3 ash.

In more recent studies, similar protein values have been reported: Sanchez et al. [14], in larvae obtained from a chicken farm, obtained an HFM with a protein level of (54%) and lipids (22%); Cheng et al. [11], in larvae produced in food waste, obtained protein values of 56.1%.

Other researchers using substrates such as manure enriched with blood and guts reported higher values: Pieterse and Pretorius [34] reported values of 60.38% crude protein, and Hwangbo et al. [35] reported values of 63.99%. Most authors have reported values higher than 50% protein and it would be associated with the type of organic substrate used for larval culture characterized by its high nitrogen content.

The lipid values present in the HFLM are on average 13.07%, which are comparable to those reported by Pieterse and Pretorius [34], and Fasakin et al. [36] with values of 14.08 and 13.65%, respectively. These authors produced larvae with a poultry manure-based feed, which is low in fiber and high in nitrogen. The lipid levels of this study are lower than those reported by Cheng et al. [11] who, based on feeding larvae with food waste, obtained 30.1% lipids, and Aniebo et al. [12] who obtained values of 25.3% based on feeding cattle blood. The substrates used by these authors would provide a greater amount of fatty acids to the larvae. A comparison of the methodology of rearing and processing of the larvae allows us to deduce that the differences reported with the other authors could be due to the feeding substrate of the larvae. Another factor that could influence the fat content of the housefly larvae is the age of the larvae, since more adult larvae have a higher lipid composition [37].

The ash content reported in this study was 10.93 ± 2.35%, similar to the 10.68% reported by Pieterse and Pretorius [34], higher than that reported by Cheng et al. [11] with values of 6.1%; Aniebo and Owen [37], with values of 7.0%; and Aniebo et al. [12] with values of 6.25%. But the ash values were lower than reported by Ogunji et al. [38], who determined that HFLM had 23.10% of ash. It has been described that the ash content varies according to the age of larval harvest, being higher when the larvae start the pupation period and it would also be associated with the diet of these since the use of organic substrates such as blood and food waste show lower ash values.

Regarding digestibility, studies with HFLM in broilers have shown variable results. Hwangbo et al. [35] reported a higher value of 98.5%. The method used to determine digestibility is important for the uniformity of the results. In this study, high levels of protein digestibility were obtained, but using in vitro techniques.

Jayanegara et al. [39] mentioned that insect meals can be used as protein supplements in ruminant feed, highlighting that all insect meals have low methane emissions in vitro. On the other hand, Makkar et al. [7] mentioned that the presence of high levels of lipids can reduce fiber digestibility in the rumen and decrease the fermentation rate, so a defatting process would be necessary before being used in ruminants.

### 4.3. Amino Acid Composition of Larval Meal

Amino acids are used by animal cells for protein synthesis, but a group of these cannot be synthesized and to form proteins they must be consumed in the diet [40]. In swine nutrition, histidine, isoleucine, leucine, lysine, methionine, phenylalanine, threonine, tryptophan and valine are described as essential amino acids. Arginine is considered an essential amino acid in the nutrition of young pigs [41]. The levels of amino acids reported in our study coincide with those reported by Cheng et al. [11], who reported an essential amino acid level of 45% for larva meals produced in food waste, highlighting the higher concentration of lysine, tyrosine, glutamic acid and aspartic acid.

Our study reports that HFLM reared in mixtures of pig and poultry manure provide adequate levels of glycine and arginine, necessary for poultry production, in addition to all the essential amino acids, highlighting their contribution of arginine, lysine, threonine and leucine. These are similar results to those reported by Aniebo et al. [12] in larvae reared on a mixture of bovine blood and wheat bran. Like traditional protein sources such as fish meal and soybean meal, HFLM reared in a mixture of manure provides all the essential amino acids, so its use in animal feed is feasible.

It has also been shown that the substitution of soybean meals by larva meal with up to 30% did not affect the digestibility coefficient of essential and nonessential amino acids in the formulated diets [9]. In egg-laying hens, it has been reported that up to 50% of the fish meal protein can be substituted without affecting egg quality [42]; in broiler chickens, an optimal inclusion rate of up to 10% of the diet is indicated, but supplementation with methionine is necessary [7].

Our study also reports a low content of Methionine and Leucine, which have the lowest amino acid scores, so supplementation with these amino acids in the formulation of balanced diets should be considered.

### 4.4. Fatty Acid Profile of Housefly Larval Meal

Edible lipids from insects are highlighted by the possibility of providing a sustainable source of nutrients for animal and human food [43]. Housefly larvae according to their diet can contain up to 36% lipids, which can be isolated and used for biodiesel preparation; additionally, the remaining defatted meal—rich in crude protein—may find a place as an invaluable protein-rich resource in the feed industry [7].

Several authors [7,34,35] have reported that the main fatty acids present in HFLM are palmitic, oleic and linoleic acids, with concentrations higher than 15% of the lipid content. Pieterse and Pretorius [34] reported that HFLM has a high content of linoleic acid with 26.25%, and linolenic acid with 2.73%, higher results than those reported in this study. These higher values of linoleic acid might be associated with the type of larval diet, as these researchers used cattle blood residues for larval feeding. In other studies, using bovine manure, a palmitic acid composition of 26.40%, oleic acid of 19.1%, linoleic acid of 17.83% and linolenic acid of 0.87% has been reported, remarkably similar to that reported in our study [44]. Using animal manure is reported to have a similar result in the fatty acid profile, with its main components being palmitic acid (29.1%), oleic acid (23.3%), palmitoleic acid (17.4%) and linoleic acid (17.2%) [45].

Hwangbo et al. [35] highlighted that the composition of HFLM can contain up to 64.11% unsaturated fatty acids when fed on egg-laying hen manure. In our study, using a mixture of swine manure and poultry manure, unsaturated fatty acids represent 53% of the lipid composition of HFLM. The effect of larval feed on their lipid composition, instead of being a disadvantage, becomes beneficial as it may allow adjusting the composition of larvae and pupae according to the nutritional needs of consumers [34].

Another viable alternative considered is the production of biodiesel from lipids obtained from housefly larvae. Biodiesel, although sustainable and renewable, faces economic limitations due to the use of human edible oils as feedstock [45]. Authors such as Zi-zhe et al. [46] reported obtaining 23.6 g of larval fat from 1 kg of poultry manure, and Yang et al. [45] obtained 21.6 g of fat from 1 kg of swine manure. Both authors pointed out that obtaining biodiesel from housefly larvae complies with ASTM D6751-10 [47], which ensures its quality and safety.

### 4.5. Evaluation of the Content of Salmonella *spp*.

Regarding the presence of *Salmonella* spp. strains in the HFLM, the samples evaluated were free of this microorganism: “Absence in 25 g”. These results coincide with those reported by Sanchez et al. [14], in which it is reported that the HFLM of larvae cultured in animal manure were free of Salmonella sp.; a similar case to the larva meal of *Hermetia illucens* [48], and it may be explained by the solar desiccation treatment given to the manure before larval culture. On the other hand, it has been determined that housefly larvae produce a peptide of the cecropin family (Mdc) with a molecular weight of 4 kDa, a molecule that has antibacterial activity against Gram-positive and Gram-negative bacteria [49]. Cecropins may also be related to the absence of *Salmonella* spp. in housefly larva meals.

This study provides valuable insights into the nutritional composition and microbiological safety of House Fly Larvae Meal (HFLM) produced under high-altitude Andean conditions. However, certain limitations must be recognized. While the chemical profile, amino acid composition and in vitro protein digestibility were assessed using validated methods, further analyses are required to confirm the suitability of HFLM as a sustainable feed ingredient. Specifically, studies addressing the presence of potential allergens, the bioavailability and absorption of nutrients in vivo, and long-term animal performance trials are crucial. These future evaluations will be critical to determine whether HFLM may be considered a viable and safe alternative protein source in animal nutrition.

## 5. Conclusions

The type of manure used to produce housefly larvae influences their performance, obtaining higher productive yields from larvae reared in a mixture of poultry manure and pig manure.

Housefly larva meal contains 56.5% crude protein, 13.07% fat, 10.93% ash, 6.77% crude fiber and an in vitro protein digestibility of 88.30%, making it a high nutritional quality feed and with an adequate balance of essential amino acids.

The absence of detectable *Salmonella* spp. indicates that housefly larva meal is microbiologically safe and suitable for inclusion in domestic animal feeding programs as a sustainable alternative protein source.

This study offers valuable information on the nutritional profile of HFLM under high-altitude Andean conditions, but further research is required to confirm its viability and safety as a protein source in animal feed.

## Figures and Tables

**Figure 1 animals-15-02054-f001:**
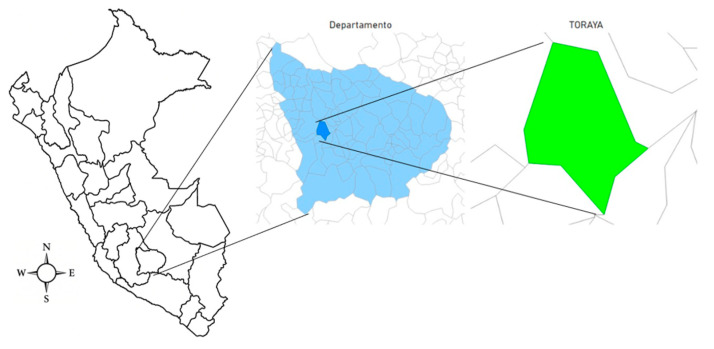
Location of the area where this study was conducted. (Source: own elaboration using Microsoft Power BI with topoJSON format).

**Table 2 animals-15-02054-t002:** Weight gain and final size of housefly larvae according to type of manure (mean ± standard error).

Type of Manure	Initial Weight (mg)	Final Weight (mg)	Total Weight Gain (mg) ^1^	Size (mm)
Pig	3.02 ± 0.01	17.70 ± 0.37 c	14.69 ± 0.37 c	8.44 ± 0.10 b
Guinea pig	3.02 ± 0.01	13.16 ± 0.71 d	10.11 ± 0.71 d	3.79 ± 0.05 e
Poultry	3.02 ± 0.01	18.40 ± 0.74 c	15.38 ± 0.74 b	5.38 ± 0.12 d
Poultry + Guinea pig	3.00 ± 0.01	13.29 ± 0.71 d	10.29 ± 0.71 d	6.89 ± 0.25 c
Pig + Guinea pig	3.02 ± 0.01	22.05 ± 0.48 b	19.03 ± 0.48 b	8.36 ± 0.27 b
Poultry + Pig	3.02 ± 0.01	27.60 ± 0.58 a	24.58 ± 0.59 a	9.47 ± 0.16 a

^1^ Different letters between rows indicate significant differences according to Tukey’s test (*p* ≤ 0.05).

**Table 3 animals-15-02054-t003:** Time of larval development in days and percentage of larval mortality according to the type of manure used for its production (mean ± standard error).

Type of Manure	Development Time (Days)	No. of Dead Larvae	Mortality (%)
Pig	10.00 ± 0.42 b	7.70 ± 1.25 ab	3.85 ± 0.62 a
Guinea pig	13.10 ± 0.43 a	4.10 ± 1.78 ab	2.05 ± 0.89 ab
Poultry	10.60 ± 0.48 b	0 c	0 c
Poultry + Guinea pig	10.50 ± 0.42 b	0.20 ± 0.20 bc	0.10 ± 0.10 bc
Pig + Guinea pig	10.30 ± 0.43 b	1.20 ± 1.00 bc	0.60 ± 0.50 bc
Poultry+ Pig	10.20 ± 0.42 b	0 c	0 c

Different letters between rows indicate significant differences according to Tukey’s test (*p* ≤ 0.05).

**Table 4 animals-15-02054-t004:** Nutritional composition and in vitro digestibility of housefly larva meal (in %).

Variable	Mean	E.E.	Lower Limit (95%)	Upper Limit (95%)
Dry matter	92.27	0.40	91.49	93.05
Soluble carbohydrates	12.03	0.78	10.50	13.56
Ethereal extract	13.07	1.34	10.44	15.70
Protein	56.50	1.04	53.76	56.24
Ashes	10.93	0.86	9.24	12.62
Crude fiber	6.77	0.35	6.08	7.46
In vitro protein digestibility	88.30	0.20	87.91	88.69
Total Energy content (MJ kg^−1^)	16.36	0.60	15.18	17.54

**Table 5 animals-15-02054-t005:** Amino acid composition of larva meal and amino acid score according to FAO (2013) [25].

Amino Acids	Composition (mg/g Protein)	Amino Acid Pattern (FAO, 2013) [25] >18 Years	Amino Acid Score * >18 Years
Aspartic acid	81.57		
Alanine	44.01		
Arginine	100.00		
Histidine	20.57	15	137.13
Isoleucine	26.65	30	88.83
Glycine	45.26		
Glutamic acid	96.60		
Lysine	59.75	45	132.77
Leucine	46.87	59	79.44
Methionine	13.06	22	59.36
Proline	32.92		
Phenylalanine	49.73	(P + T) 38	257.02
Tyrosine	47.94		
Serine	38.64		
Threonine	83.72	23	364.00
Tryptophan	8.77	6	146.16
Valine	38.28	39	98.15

* Amino acid score applies only to essential amino acids.

**Table 6 animals-15-02054-t006:** Lipid profile of larva meal grown in a mixture of pig and poultry manure (in %).

Fatty Acid		Mean	E.E.	Lower Limit (95%)	Upper Limit (95%)
C12:0	Lauric acid	0.26	0.01	0.24	0.28
C14:0	Myristic acid	4.13	0.11	3.91	4.35
C14:1	Myristoleic acid	0.38	0.01	0.36	0.40
C15:0	Pentadecanoic acid	2.63	0.05	2.53	2.73
C16:0	Palmitic acid	29.34	0.06	29.22	29.46
C16:1	Palmitoleic acid	21.65	0.07	21.51	21.79
C17:0	Heptadecanoic acid	1.11	0.01	1.09	1.13
C18:0	Stearic acid	8.66	0.06	8.54	8.78
C18:1n-9	Cis-9 oleic acid	26.53	0.12	26.29	26.76
C18:2	Linoleic acid	4.19	0.13	3.94	4.44
C20:0	Arachidic acid	0.17	0.01	0.15	0.19
C18:3n-3	Linolenic acid	0.26	0.01	0.24	0.28
C21:0	Heneicosanoic acid	0.69	0.08	0.53	0.85

## Data Availability

The raw data supporting the conclusions of this article will be made available by the authors on request.

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
