# Peer review of "Productive Yield, Composition and Nutritional Value of Housefly Larva Meal Reared in High-Altitude Andean Zones of Peru"

_animals, 2025, doi:10.3390/ani15142054_

Round 1

Reviewer 1 Report

Comments and Suggestions for Authors

This study investigates the production, composition and nutritional value of meal from housefly larvae raised in the Andean areas of Peru. The results indicate that these larvae can be a viable and nutritious alternative for animal feed. 
The nutritional composition includes essential amino acids, especially lysine and methionine.
The production of larvae can contribute to the reduction of organic waste and sustainability in agriculture and the use of fly larvae for the decomposition of organic waste can result in a significant reduction in methane emissions.

S
The analysis of the composition of the larvae includes the evaluation of digestibility and the presence of essential nutrients. The digestibility of animal protein was evaluated using methods such as pepsin digestion. The composition of the larvae includes proteins, fibers, ash and carbohydrates, with variations depending on the larvae's diet. The research also discusses the production of biodiesel from larval waste, showing the versatility of this resource.

The article is simple and clear. Nevertheless, it brings novel and valuables information. The experimental design is sound and the methodology employed is appropriated and well established. The objectives are clearly stated and this work deals with a relevant issue, shedding light on approaches to promoting the circular economy by optimizing the destination of mannure and animal protein production.. The findings are interesting and compelling. The original results fall into the scope of the Journal and they could be published with minor revision. 

Comments on the Quality of English Language

The English used is quite good, but it deserves to be reviewed by a native speaker of the language.

Author Response

We sincerely thank the reviewer for their thoughtful and constructive comments on our manuscript.

The manuscript has been thoroughly revised by a specialist with proven experience in the translation and editing of scientific manuscripts, to ensure that the language meets the standards expected for publication.

Reviewer 2 Report

Comments and Suggestions for Authors

Abstract 

The abstract should not conclude that housefly larvae meal is suitable for use as an ingredient, as this is premature. First, analyses regarding allergens, nutrient absorption, and bioavailability must be conducted to assess its viability. Additionally, a study involving animals should be carried out to verify its effects. Therefore, while it can be presented as a hypothesis, further testing is necessary before reaching a definitive conclusion.

Figure 1

Please specify the software used to create Figure 1.

Methods: Determine if there is a government agency in the city that can authorize large-scale fly trapping. In some countries, permits are required to conduct studies involving wild insects.

Explain why the microorganism Salmonella was chosen for analysis. Are there no other pathogenic bacteria that affect animals?

The modified AOAC 971.09 method needs detailed explanation, as it has been revised and is among the most important analyses. This information should be included in the abstract, as it is currently missing.

It is essential to specify the number of samples and repetitions for each analysis.

Discussion

A paragraph should discuss the study's limitations, indicating that further analyses (including allergens, bioavailability, absorption, etc.) and animal trials are necessary to determine if the flour can be a viable alternative.

Conclusions

Enhancing the conclusions requires extensive data; this section can be expanded.

Author Response

We sincerely appreciate the detailed review and the constructive comments provided on our manuscript. We believe that your observations have been very valuable in enhancing the scientific quality of our work. Below, we present our point-by-point responses to each of the suggestions and comments made.

Comments 1: Abstract. The abstract should not conclude that housefly larvae meal is suitable for use as an ingredient, as this is premature. First, analyses regarding allergens, nutrient absorption, and bioavailability must be conducted to assess its viability. Additionally, a study involving animals should be carried out to verify its effects. Therefore, while it can be presented as a hypothesis, further testing is necessary before reaching a definitive conclusion.

Response 1: We appreciate the reviewer’s valuable observation regarding the conclusion in the abstract. As recommended, we have revised the final sentence of the abstract to avoid making a definitive statement about the suitability of housefly larvae meal as an animal feed ingredient. Instead, we now present it as a potential alternative that warrants further investigation. The revised version acknowledges the need for additional studies, particularly those evaluating allergenicity, nutrient bioavailability, and in vivo performance, before confirming its viability for practical application. These changes have been made on page 1, lines 34 to 37 of the revised manuscript.

Comments 2: Figure 1. Please specify the software used to create Figure 1.

Response 2: We thank the reviewer for this observation. As requested, we have specified the software used to create Figure 1 in the figure caption. The caption now clearly states that the map was generated using Microsoft Power BI with TopoJSON format. This clarification has been included on page 3, lines 106–107 of the revised manuscript.

Comments 3: Methods: Determine if there is a government agency in the city that can authorize large-scale fly trapping. In some countries, permits are required to conduct studies involving wild insects.

Response 3: We appreciate the reviewer’s concern regarding regulatory approval for fly trapping. In response, we have added a clarification in the Methods section indicating that, in Peru, there is currently no specific governmental institution or legislation that prohibits or regulates the use of house flies (Musca domestica) for research purposes. Therefore, no special permits were required to carry out this study. This information has been incorporated on page 3, lines 113–116 of the revised manuscript.

Comments 4: Explain why the microorganism Salmonella was chosen for analysis. Are there no other pathogenic bacteria that affect animals?

Response 4: We appreciate the reviewer’s insightful question regarding the selection of Salmonella spp. for microbiological analysis. In Peru, national food safety regulations—specifically the Technical Health Standard NTS N° 071-MINSA/DIGESA-V.01—establish that food and feed products must be free from Salmonella contamination. This microorganism is considered a critical indicator of sanitary quality and safety, both in human food and animal feed systems. While other pathogenic bacteria may also be relevant, Salmonella is prioritized in national regulations due to its zoonotic potential and its impact on public health. Therefore, it was selected as the reference microorganism for the microbiological assessment of the larvae meal. This clarification has been added to the Methods section on page 5, lines 181–185 of the revised manuscript.

Comments 5: The modified AOAC 971.09 method needs detailed explanation, as it has been revised and is among the most important analyses. This information should be included in the abstract, as it is currently missing.

Response 5: We thank the reviewer for this important observation. After reviewing the methodology, we confirm that in vitro protein digestibility was determined using the method described by Becker (1961), which simulates gastric digestion using pepsin in an acid medium. This method is widely recognized for evaluating the digestibility of feed and forage protein sources. As suggested, we have provided a detailed description of the procedure in the Methods section, including incubation conditions, sample preparation, and the calculation formula. These details can be found on page 5, lines 188 and 195 of the revised manuscript.

Commenrs 6: It is essential to specify the number of samples and repetitions for each analysis.

Response 6: We thank the reviewer for this important observation. In response, we have specified the number of samples used for each analysis performed in the study. This information has been incorporated into the Methods section to improve transparency and reproducibility. The corresponding changes can be found on page 5, lines 169–173, 176, 179, 188, and 199 of the revised manuscript.

Comments 7: Discussion: A paragraph should discuss the study's limitations, indicating that further analyses (including allergens, bioavailability, absorption, etc.) and animal trials are necessary to determine if the flour can be a viable alternative.

Response 7: We appreciate the reviewer’s recommendation to include a discussion of the study’s limitations. In response, we have added a paragraph in the Discussion section acknowledging the current limitations of our research and emphasizing the need for further analyses—such as assessments of allergenicity, nutrient bioavailability, absorption, and in vivo animal trials—to comprehensively evaluate the viability of housefly larvae meal as a sustainable feed ingredient. These additions have been made on page 11, lines 429–437 of the revised manuscript.

Comments 8: Conclusions: Enhancing the conclusions requires extensive data; this section can be expanded.

Response 8: We thank the reviewer for the suggestion to strengthen the conclusions. In response, we have expanded the Conclusions section to provide a more comprehensive synthesis of the main findings of the study, highlighting the nutritional value, microbiological safety, and potential of housefly larvae meal as a sustainable protein source. The revised conclusion also reflects the limitations and the need for further research, in line with the discussion.

Round 2

Reviewer 2 Report

Comments and Suggestions for Authors

All requested changes have been made